# Communication, Computing, and Caching Trade-Off in VR Networks

Yuqing Feng [1], Dongyu Wang [2] and Yanzhao Hou [3],*

[1] School of International Studies, Beijing University of Posts and Telecommunications, Beijing 100876, China
[2] The Key Laboratory of Universal Wireless Communication, Ministry of Education, Beijing University of Posts and Telecommunications, Beijing 100876, China
[3] National Engineering Research Center for Mobile Network Technologies, Beijing University of Posts and Telecommunications, Beijing 100876, China
* Correspondence: houyanzhao@bupt.edu.cn

**Abstract:** As technology continues to advance, virtual reality (VR) video services are able to provide an increasingly realistic video experience. VR applications are limited, since the creation of an immersive experience requires processing and delivery of incredibly huge amounts of data. A potential technique to decrease the operation time for VR, as well as its energy use, is mobile edge computing (MEC). In this study, we develop a VR network in which several MEC servers can supply field-of-view (FOV) files to a VR device in order to satisfy the transmission requirements of VR video service and improve the quality of the experience. In this way, the projection process from 2D FOV to 3D FOV and the cached data is possible on an MEC server or a VR device. A cooperative computational offloading and caching strategy is developed as a decision matrix to reduce transmission requirements based on the service time constraint requirement. The VR video service mechanism is examined through the decision matrix. The trade-off between communication, caching, and computation (3C trade-off) is further implemented by means of a closed equation for the decision matrix. Results from simulations show that the suggested technique can perform close to optimally compared to alternative opposing methods.

**Keywords:** MEC; wireless caching; VR video service; 3C trade-off





## 1. Introduction

Computer simulation technology called virtual reality (VR) enables the construction and enjoyment of a virtual world by creating a simulated environment in which the user is fully immersed [1,2]. VR uses real-life data to create phenomena that humans can perceive through electronic signals created by computer technology, combined with a variety of output devices, which can be actual objects in reality or substances that are invisible to the human eye, expressed through three-dimensional models. One of the key uses for 6G networks is anticipated to be VR [3]. Moreover, a gigabyte of data must be delivered with an ultra-low latency of about 20 ms for VR video to be fully immersive [4]. This necessitates an extremely high transmission rate and places a heavy weight on the backhaul network.

It should be noted that various user fields of views (FOVs), which are determined by trackers in VR devices, may change when users view the same 360 degree film. In addition, the FOV transmission scheme introduced in [5] mainly focuses on high-quality images transmitted in the determined field of view, which is an effective way to reduce the total amount of data transmitted. This is in sharp contrast to the full field-of-view transmission scheme that requires the transmission of all video tiles, for example, Huawei's Tile Wise transmission scheme and Facebook's Pyramid Projection transmission scheme. FOV transmission is the main subject of this article, and the FOV transmission strategy is chosen based on its advantages in reducing transmission burden and greatly improving user quality of experience (QoE). Accordingly, a conventional VR framework (Figure 1 [6]) should be

introduced. The steps involved in creating VR video are as follows: (1) A combination of videos recorded by a multicamera array produces a spherical movie. (2) The spherical video is unfolded to produce a 2D video using equirectangular projection. (3) Extraction: The FOV corresponding to the point of view from a 2D video is obtained. (4) Projection: This 2D FOV is converted into a 3D FOV. (5) Extraction, equirectangular projection and render stitching are the first three preprocessing stages and should be done conducted offline on a cloud server with centralized computing power, as they combine many periods of 360 degree footage or require the entire 360 degree video as input. There is no doubt that the tracking and rendering process is already complete for VR devices. The difficulty in mitigating the heavy transmission requirements of wireless backhaul connections for VR video lies in completing the projection process.

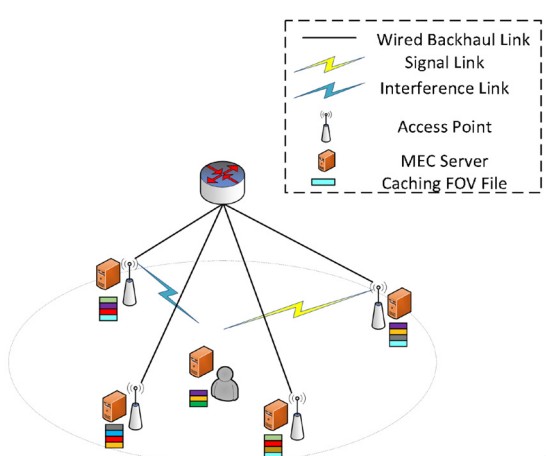

**Figure 1.** VR network architecture.

One of the main ways to implement VR video services is traditional mobile edge computing (MEC) architecture [7], which can utilize the caching and processing capabilities of edge networks and be combined with wireless transmission networks. First, selected FOV files are actively stored in the edge network by the caching function during off-peak periods in preparation for future demand, making full use of the resources (computational and caching power) of the edge network (both VR devices and MEC servers) [8]. Second, by utilizing the increasing computing capability of edge an central processing unit (CPU), the projection technique can reduce service times [9]. In this paper, we examine the VR video service mechanism in the MEC network architecture in order to reduce transmission needs while fulfilling the strict service time constraint. The best strategy to utilize the computing and caching capabilities of an edge network is then decided. This is referred to as the communication, caching, and computation trade-off (3C trade-off).

### 1.1. Related Works

Many academics and developers at high-tech companies are currently working hard to make VR services mainstream. Large amounts of data are required to provide the foundation for the creation of VR situations in order to increase the realism and immersion of the experiences. As each user can only experience VR from one perspective at a time, FOV is supplied selectively rather than for the entire VR video [5]. The authors of [10] developed a VR QoE evaluation system based on VR transmission parameters; however, the model did not make use of an MEC network. Instead, they constructed an upgraded neural network model based on cognitive neurological properties.

In terms of VR video services, in order for an MEC network realize its full capabilities the authors of [11], described a QoE gain-based precaching method and VR video distribution system. A hierarchical collaborative caching problem was presented in [12] in the form of an approach that reduces transmission latency, and a VR caching system

based on visual synthesis was created. The MaxMinDistance caching algorithm has a significant role in solving the NP-hard problem. In order to significantly enhance the quality of experience for VR video services, the integrated processing and caching capabilities of edge networks were identified as key enablers. According to the computation offloading approach and collaborative caching allocation set out in [13], the MEC servers cooperated to carry out computation workloads and data caching. The core 3C trade-off problem for VR services was explored in the context of optimizing VR video services created in cache-based supported cooperative networks [14]. The authors integrated challenges associated with decision making, communication, and caching. In order to maximize the average tolerable latency, a mobile VR framework was proposed [15]. This framework works primarily by caching a portion of VR movies in advance and running a specialized handler at the edge. The authors of [16] suggested a joint optimization algorithm based on an asynchronous dominant actor–critic in order to obtain an ideal viewport rendering offload and transmission power control approach. The program was intended to produce a high-quality immersive VR video experience while lowering the long-term energy consumption of the MEC system.

Furthermore, the authors of [17,18] utilized the caching and computational capabilities of either a VR device or an MEC server; however, there is still room for improvement in the synergy between the two, and an important issue is that associated with 2D FOV files. One option is to compute the route locally, with the MEC server taking care of caching the 2D FOV, and the other is to cache and compute the 2D FOV to meet the requirements. The applicability of the 3C collaboration model was analyzed in [19], in which the authors defined collaboration tools in collaborative virtual reality applications. The authors of [20,21] proposed a proactive 3C resource allocation method to provide low-latency VR video to clients. However, in [17–21], the authors did not present a concrete implementation of the VR framework. A standard VR framework was presented in [6], but its main computational power was used to reduce the load of communication. In this paper, we propose a cooperative caching and computational offloading technique that employs the caching and computational power of MEC servers and VR devices in a VR network. Through problem analysis, we identify the link between decision making and the 3C trade-off.

### 1.2. Contributions and Outcomes

3C trade-offs in VR video services are covered in this article. Via a VR network, several MEC servers can transmit FOV files to the VR device for VR video services. This approach allows projection calculations or video file caching to be performed on the MEC server, significantly reducing the transfer requirements of the VR service and increasing transfer efficiency. Computing offloading and the cooperative caching approach are presented using a decision matrix. An optimization problem based on a choice matrix is then built to minimize the mean of the lowest necessary transmission rate, taking into account caching and computing capacity limits. The service mechanism of VR is analyzed using the various values of the matrix. This analysis of choice characteristics, which are connected to 3C trade-off, yields a closed form of the decision matrix.

The rest of the article is structured as follows. The system model, which consists of a VR network architecture, models for cooperative caching and computational offloading, as well as a service mechanism, is presented in Part II of the article. In Part III, we define the system issue in a scenario including many videos and suggest an optimization technique. The simulation findings are presented in Part IV of the study, and we present our conclusions in Part V.

## 2. System Model

In this section, we describe a network that uses cooperative caching and computation offloading, as well as a service mechanism for handling random requests.

### 2.1. Network Framework

A VR device with caching and computing capabilities is set up. Suppose that there are *K* uniformly distributed single antenna access points (APs) in a circular area centred on the VR device, with a radius of propagation of the APs of $R_M$. $\Phi_A$ denotes the set of APs. At APs, the MEC servers possess a cache of the same size ($Q_M S$ (bit)), as well as a cache of size $Q_D S$ (bit) at the VR device, where $S$ (bit) is the size of the file of each viewpoint. In general, $Q_D < Q_M$, and all values are an integer.

An FOV-based VR movie library is offered, where users may access the 3D FOV and 2D FOV of each film, taking into account both the distinction between 3D and 2D FOV and the popularity of the videos. The video library ($\mathcal{V} = \{1, 2, \dots, N\}$) contains $N = |\mathcal{V}|$. The same-sized VR videos are uploaded to a cloud server (CS) and ordered in decreasing order by popularity, leading to the rule that more well-liked videos are rated with lower indices. In accordance with Zipf's law, it is believed that the popularity of a video represents the possibility that it will be randomly requested over the whole network.

$$p_{v_n} = \frac{n^{-\alpha}}{\sum_{l=1}^{N} l^{-\alpha}}, n \in \mathcal{V} \tag{1}$$

where $\alpha$ is the skewness parameter characterizing the concentration of video requests. In addition, each video is assumed to have $M$ viewpoints based on the extraction process, and the space of viewpoints is denoted as $\mathcal{P} = \{1, \dots, M\}$. When VR users navigate panoramic scenes, the frequency of tracking viewpoint $m$ can be captured by the probability ($p_{p_m}$) of the VR device requesting viewpoint $m \in \mathcal{P}$. We consider uniform distribution, i.e., $p_{p_m} = \frac{1}{M}$, as the viewpoint popularity. It is assumed that the video popularity and the viewpoint popularity are independent without loss of generality. Hence, the requesting probability of the viewpoint $m$ in video $n$ can be modeled as

$$p_{n,m} = p_{v_n} p_{p_m} = \frac{n^{-\alpha}}{M \sum_{l=1}^{N} l^{-\alpha}}, n \in \mathcal{V}; m \in \mathcal{P} \tag{2}$$

Because the user's right and left eyes have two exposed perspectives, the projection procedure for each viewpoint needs to be computed twice in order to generate stereo vision. $\widetilde{S}$ represents the 3D FOV of each perspective. Each 2D FOV file is at least twice as small as a 3D FOV file in terms of size [4]. Therefore, $\rho = \frac{\widetilde{S}}{S} \geq 2$. The projection calculation of the transformation from 2D FOV to 3D FOV of every perspective in any video can be represented by three parameters $\left(S, \widetilde{S}, \eta\right)$, where $\eta$ denotes the quantity of calculation cycles (cycle/bit) needed to analyze a bit of data.

### 2.2. Computing–Offloading and Cooperative Caching Strategy

The cooperation mode of the MEC server and the VR device storing FOV files is 0-1 cache in the cache placement phase, which indicates whether the FOV file is stored completely. Since all *K* MEC servers can serve VR devices when idle, the caching rule is 'maximum diversity content'. Therefore, the cache resource of the *K* MEC server can be regarded as a cache space of $K Q_M S$, and the FOV files can be stored differently, which is possible through network function virtualization (NFV). The vector $\mathbf{c}_{n,m}$ represents the caching decision for the 2D FOV, which is the $m$ viewpoint in video $n$, where

$$\mathbf{c}_{n,m} \begin{cases} [1\ 0\ 0], \textit{No Caching} \\ [0\ 1\ 0], \textit{Local Caching} \\ [0\ 0\ 1], \textit{MEC Caching} \end{cases} \tag{3}$$

where $\mathbf{c}_{n,m} = [0\ 0\ 1]$ means that the FOV can be found by the MEC server, and $\mathbf{c}_{n,m} = [0\ 1\ 0]$ means that the video files can be found at the VR device; if the FOV cannot be found by the

above two methods, then the file cannot be found on the MEC server or the VR device. The cache representation of the video files for viewpoint $m$ in video $n$ is e $\tilde{\mathbf{c}}_{n,m}$.

$$\tilde{\mathbf{c}}_{n,m}\begin{cases} [1\ 0\ 0], \textit{No Caching} \\ [0\ 1\ 0], \textit{Local Caching} \\ [0\ 0\ 1], \textit{MEC Caching} \end{cases} \tag{4}$$

The choice to offload the projection process of perspective $m$ in video $n$ on the MEC server or VR device is represented by the vector $\mathbf{d}_{n,m}$, taking into account the computational capability of VR devices and MEC servers.

$$\mathbf{d}_{n,m}\begin{cases} [1\ 0\ 0], \textit{No Computing} \\ [0\ 1\ 0], \textit{Local Computing} \\ [0\ 0\ 1], \textit{MEC Computing} \end{cases} \tag{5}$$

When the projection process can be calculated in $K$ MEC servers, $\mathbf{d}_{n,m} = [0\ 0\ 1]$. On a VR device, $\mathbf{d}_{n,m} = [0\ 1\ 0]$ indicates that the corresponding view from the known file to the desired 3D file can be generated. If none of the above conditions is satisfied, neither the MEC server nor the VR device has computing power.

Combining the computing–offloading decisions and joint caching of viewpoint $m$ in video $n$ yields a corresponding $3 \times 3$ decision matrix: $\mathbf{D}^{n,m} = \left[ \mathbf{c}_{n,m}{}^T\ \tilde{\mathbf{c}}_{n,m}{}^T\ \mathbf{d}_{n,m}{}^T \right]$. A decision matrix set ($\mathcal{D}$) is made up of the decision matrices for each perspective in each VR video. Due to the limited cache capacities of the MEC server and the VR device,

$$\sum_{n=1}^{N} \sum_{m=1}^{M} \mathbf{D}_{2,1}^{n,m} + \rho \mathbf{D}_{2,2}^{n,m} \leq Q_D \tag{6}$$

$$\sum_{n=1}^{N} \sum_{m=1}^{M} \mathbf{D}_{3,1}^{n,m} + \rho \mathbf{D}_{3,2}^{n,m} \leq K Q_M \tag{7}$$

where $f_M$ and $f_D$ (cycle/s) represent the MEC server and VR device running at two different given CPU cycle frequencies. Normally, $f_M > f_D$. $\kappa f_D{}^2$ represents the energy consumed to calculate a period of $f_U$ on a VR device, where $\kappa$ is the CPU energy efficiency parameter of the MEC server and the VR device. The average power consumption limitation ($P_D(\mathrm{W})$) is taken into account in order to guarantee that the average time for users to view VR video is more than or equal to a specific amount of time and assuming that the energy capacity of VR devices is constant and restricted.

$$\frac{\kappa f_D{}^2 S \eta \sum_{n=1}^{N} p_{v_n} \sum_{m=1}^{M} \mathbf{D}_{2,3}^{n,m}}{M \tau} \leq P_D \tag{8}$$

According to (8), when a VR movie is randomly requested, the maximum value of the projection that can be calculated on the device is $\lfloor \frac{P_D M \tau}{\kappa f_D{}^2 S \eta} \rfloor$, where $\lfloor \rfloor$ is the floor operation.

### 2.3. Service Mechanism

In order to avoid physical discomfort, the VR device requires that each request be completed within a time limit of seconds. Due to the wired backhaul network's limited transmission capacity between CS and MEC servers, it is anticipated that the transmission time over the link would be much longer while downloading 3D/2D FOV data from CS. Each of the seven routes described below that satisfy the service time constraint yields a minimum transmission rate and can satisfy the request for viewpoint $m$ in video $n$, which is designated as $R_{n,m}$ (bit/s), in accordance with the decision matrix ($\mathbf{D}^{n,m}$).

(1)　3D Caching Locally (3CL)

If $\mathbf{D}_{2,2}^{n,m} = 1$, without additional processing power or communication, it is possible to locally retrieve the 3D FOV of the appropriate perspective. In this case, the service time is insignificant, and $R_{n,m} = 0$ is the lowest necessary transmission rate.

(2) 2D Caching—Computing Locally (2CCL)

If $\left(1 - \mathbf{D}_{2,2}^{n,m}\right)\mathbf{D}_{2,1}^{n,m}\mathbf{D}_{2,3}^{n,m} = 1$, the VR device locally captures the 2D FOV of the relevant perspective before converting it to a 3D FOV using the onboard processor. The necessary minimum transmission rate is $R_{n,m} = 0$, and the service duration is $\frac{S\eta}{f_D}$ (second). Because the processing power of the VR device increases considerably, it is predicted that $\frac{S\eta}{f_D} \leq \tau$ or, in VR devices, the projection process is computed and can be completed within the service time restriction.

(3) 2D Computing Locally (2CL)

If $\left(1 - \mathbf{D}_{2,2}^{n,m}\right)\left(1 - \mathbf{D}_{2,1}^{n,m}\right)\mathbf{D}_{2,3}^{n,m} = 1$, the device projects the video files of the relevant perspective into the 3D FOV using the local CPU after downloading it from the servers or CS. When $\mathbf{c}_{n,m} = [0\ 0\ 1]$, i.e., $\left(1 - \mathbf{D}_{2,2}^{n,m}\right)\mathbf{D}_{3,1}^{n,m}\mathbf{D}_{2,3}^{n,m} = 1$, the closest MEC server will have the relevant 2D FOV file available. 2CL-FM is the route's official name. When $\mathbf{c}_{n,m} = [1\ 0\ 0]$ and $\tilde{\mathbf{c}}_{n,m} = [1\ 0\ 0]$, i.e., $\mathbf{D}_{1,1}^{n,m}\mathbf{D}_{1,2}^{n,m}\mathbf{D}_{2,3}^{n,m} = 1$, the relay station that obtains the matching 2D FOV from the CS and sends it to the device is the MEC server that is nearest to the device. Its designation is 2CL-FS. $\frac{S}{R_{n,m}} + \frac{S\eta}{f_D} + 10\left(1 - \mathbf{D}_{3,1}^{n,m}\right)\tau$ indicates the amount of time spent serving through the 2CL route, where $10\tau$ is the significantly harsher penalty for obtaining FOV files from the CS. The service time for the 2CL-FM route is defined as $\frac{S}{R_{n,m}} + \frac{S\eta}{f_D}$, and given the restriction on service time, $\frac{S}{R_{n,m}} + \frac{S\eta}{f_D} \leq \tau$. Hence, it is possible to achieve the minimal necessary 2CL-FM transmission rate, i.e., $R_{n,m} = R^{LC-FM} \triangleq \frac{S}{\tau - \frac{S\eta}{f_D}}$.

$10\tau + \frac{S}{R_{n,m}} + \frac{S\eta}{f_D}$ denotes the service time under the routing of 2CL-FS. Given that the final two components in the formula presented above are all non-negative, $10\tau + \frac{S}{R_{n,m}} + \frac{S\eta}{f_D} \gg \tau$, i.e., the service time requirement is not met by the 2CL-FS route.

(4) 3D Caching MEC (3CM)

If $\mathbf{D}_{3,2}^{n,m}(1 - \mathbf{D}_{2,1}^{n,m}\mathbf{D}_{2,3}^{n,m}) = 1$, the FOV of the matched viewpoint is downloaded by the VR device from the nearest server that has the file. Service time is defined as $\frac{\tilde{S}}{R_{n,m}}$. With respect to the service time limitation $\frac{\tilde{S}}{R_{n,m}} \leq \tau$, the necessary minimum transmission rate is $R_{n,m} = R^{M3C} \triangleq \frac{\tilde{S}}{\tau}$.

(5) 2D Caching-Computing MEC (2CCM)

If $\mathbf{D}_{3,1}^{n,m}\mathbf{D}_{1,2}^{n,m}\mathbf{D}_{3,3}^{n,m} = 1$, the 2D FOV files are converted to a 3D FOV by the MEC, which then sends the 3D FOV files to the VR device. $\frac{S\eta}{f_M} + \frac{\tilde{S}}{R_{n,m}}$ is the service time under the constraint $\frac{S\eta}{f_M} + \frac{\tilde{S}}{R_{n,m}} \leq \tau$, and the minimum required transmission rate is $R_{n,m} = R^{MC2C} \triangleq \frac{\tilde{S}}{\tau - \frac{S\eta}{f_M}}$.

(6) 2D Computing MEC (2CM)

If $(1 - \mathbf{D}_{3,1}^{n,m})\mathbf{D}_{1,2}^{n,m}\mathbf{D}_{3,3}^{n,m} = 1$, the appropriate 2D FOV file is downloaded from the CS to the nearest MEC server, which then uses MEC CPU to project it into a 3D FOV file that is subsequently sent to the VR device. $10\tau + \frac{S\eta}{f_M} + \frac{\tilde{S}}{R_{n,m}}$ is the service time. Since the latter two terms in the above formular are all non-negative, $10\tau + \frac{S\eta}{f_M} + \frac{\tilde{S}}{R_{n,m}} \gg \tau$, i.e., the service time constraint is not satisfied by the 2CM route.

(7) CS Downloading (CD)

If $\mathbf{D}_{1,2}^{n,m}\mathbf{D}_{1,3}^{n,m} = 1$, instead of additional calculations by the designated MEC server closest to the VR device, the matching 3D FOV file is downloaded directly from the CS and transferred to the VR device. $20\tau + \frac{\tilde{S}}{R_{n,m}}$ is the service time. Furthermore, $20\tau + \frac{\tilde{S}}{R_{n,m}} \gg \tau$, i.e., the service time constraint not met by CD.

The service time corresponding to one of the three routes (2CL-FS, 2CM, or CD) is extremely long due to the significant transmission penalty in these routes. $R_{n,m} = R^P \triangleq \varsigma\frac{\widetilde{S}}{\tau}$ are the transmission rates that we uniformly assume for the three methods, where $\varsigma \gg 1$.

## 3. Problem Construction and Solution

In this section, we propose a system problem and the corresponding optimization method.

### 3.1. Problem Definition

For matrix $\mathbf{D}^{n,m}$, the mean of the minimum required transmission rates are shown in formulas (9) and (10)

$$
\begin{aligned}
R = \sum_{n=1}^{N} \sum_{m=1}^{M} p_{n,m} & \left\{ \left(1 - \mathbf{D}_{2,2}^{n,m}\right)\mathbf{D}_{3,1}^{n,m}\mathbf{D}_{2,3}^{n,m}R^{LC-FM} + \mathbf{D}_{3,2}^{n,m}(1 - \mathbf{D}_{2,3}^{n,m} + \mathbf{D}_{1,1}^{n,m}\mathbf{D}_{2,3}^{n,m})R^{M3C} \right. \\
& \left. + \mathbf{D}_{3,1}^{n,m}\mathbf{D}_{1,2}^{n,m}\mathbf{D}_{3,3}^{n,m}R^{MC2C} + \mathbf{D}_{1,2}^{n,m}\left(\mathbf{D}_{1,3}^{n,m} + \mathbf{D}_{3,3}^{n,m} + \mathbf{D}_{1,1}^{n,m}\mathbf{D}_{2,3}^{n,m} - \mathbf{D}_{3,3}^{n,m}\mathbf{D}_{3,1}^{n,m}\right)R^P \right\}
\end{aligned}
\tag{9}
$$

$$
\begin{aligned}
R = \sum_{n=1}^{N} \sum_{m=1}^{M} p_{n,m} & \left\{ \mathbf{D}_{3,1}^{n,m}\mathbf{D}_{1,2}^{n,m}\mathbf{D}_{2,3}^{n,m}R^{LC-FM} + \mathbf{D}_{3,2}^{n,m}(1 - \mathbf{D}_{2,1}^{n,m}\mathbf{D}_{2,3}^{n,m})R^{M3C} + \mathbf{D}_{3,1}^{n,m}\mathbf{D}_{1,2}^{n,m}\mathbf{D}_{3,3}^{n,m}R^{MC2C} \right. \\
& \left. + \mathbf{D}_{1,2}^{n,m}\left(\mathbf{D}_{1,3}^{n,m} + \mathbf{D}_{3,3}^{n,m} + \mathbf{D}_{1,1}^{n,m}\mathbf{D}_{2,3}^{n,m} - \mathbf{D}_{3,3}^{n,m}\mathbf{D}_{3,1}^{n,m}\right)R^P \right\}
\end{aligned}
\tag{10}
$$

where $f_D \geq \frac{\rho S\eta}{(\rho-1)\tau}$ and $f_D < \frac{\rho S\eta}{(\rho-1)\tau}$. For the decision matrix set $(\mathcal{D})$, the specific system problem can be expressed as:

Problem 1.

$$
\min_{\mathcal{D}} R
$$

$$
s.t. \sum_{n=1}^{N} \sum_{m=1}^{M} \mathbf{D}_{2,1}^{n,m} + \rho\mathbf{D}_{2,2}^{n,m} \leq Q_D
\tag{11}
$$

$$
\sum_{n=1}^{N} \sum_{m=1}^{M} \mathbf{D}_{3,1}^{n,m} + \rho\mathbf{D}_{3,2}^{n,m} \leq KQ_M
\tag{12}
$$

$$
\sum_{n=1}^{N} p_{v_n} \sum_{m=1}^{M} \mathbf{D}_{2,3}^{n,m} \leq \frac{P_D M\tau}{\kappa f_D^2 S\eta}
\tag{13}
$$

$$
\mathbf{D}_{i,j}^{n,m} \in \{0,1\}, \forall i,j \in \{1,2,3\}; \sum_{i=1}^{3} \mathbf{D}_{i,j}^{n,m} = 1, \forall j \in \{1,2,3\}
\tag{14}
$$

$$
n \in \mathcal{V}; m \in \mathcal{P}
$$

### 3.2. DDPG-Based Optimization Method

The optimization of Problem 1 can be expressed as a Markov decision process (MDP) problem. Several elements make up an MDP $\langle S, A, P, R, \gamma \rangle$.

- State: $S_t = \mathcal{D}(t)$ denotes the state $(S)$ at time $t$, which is the value of the decision matrix set;
- Action: We define $A_t = \mathbf{D}_{i,j}^{n,m} - \rho\frac{\partial R}{\partial \mathbf{D}_{i,j}^{n,m}}$, where $\rho < 1$;
- State transition probability: $P(S_{t+1}|S_t, A_t)$ denotes the likelihood that the current state and its behavior will continue in the future;
- Reward: The agent receives an immediate reward $R(t)$ at the end moment of each $t$;
- Discount factor: We define $\gamma \in [0, 1]$ as the discount factor. Therefore, the long-term reward of the system can be expressed as $\overline{G}_\pi = \frac{1}{T}\mathbb{E}_\pi\left[\sum_{t=0}^{T} \gamma^t R_t \Big| S_0\right]$.

Obviously, the state and action space of the scene are continuous, which makes table-based value function methods such as Q-learning or DQN not applicable. We want to build a mapper from a continuous state to a continuous action as a strategy learning model. We therefore address this MDP problem using the deep deterministic policy gradient (DDPG) approach. DDPG is an actor–critic framework-based, policy-based algorithm. To decrease

the correlation between subsequent samples and improve learning stability, experience replay and target networks are also utilized in combination with DDPG. The input consists of the main value network ($\xi$), the main policy network ($\mu$), the target value network ($\xi\prime$), and the target policy network ($\mu\prime$), which are initialized using the parameters $\theta_\mu$, $\theta_{\mu\prime}$, $\theta_\xi$, and $\theta_{\xi\prime}$ respectively. We also set an appropriate value for the discount factor ($\gamma$), soft renewal coefficient ($\tau$), the sample size of batch gradient descent ($m$), target network parameter update frequency ($L$), and the maximum number of iterations ($T$). The whole algorithm is divided into two periods: the first, in which experience is accumulated, and the second, in which model training is completed. In the first phase, training data are generated using a policy network with random initialization of parameters. Each tuple $\{S_t,\ A_t, R_t,\ S_{t+1}\}$ is stored in the replay buffer. Once sufficient sample data are available, the model training phase begins. During each iteration, a minibatch is randomly drawn from the sampled data; then, the main strategy and value network are trained alternately. In this paper, we use $\phi(S)$ to denote the feature vector of S. For the main value network, network optimization is based on minimizing the gradient of the mean square error.

$$J(\theta_\xi) = \frac{1}{m} \sum_{j=1}^{m} (y_j - Q(\phi(S_j), A_j, \theta_\xi)^2 \tag{15}$$

where $y_j = R_j + \gamma Q\prime\left(\phi(S\prime_j), \pi_{\theta_{\xi\prime}}(\phi(S\prime_j)), \theta_{\xi\prime}\right)$.

In order to further improve the resource utilization, we optimize the parameters of the main policy network [22]:

$$\nabla J(\theta_\mu) = \frac{1}{m} \sum_{j=1}^{m} \nabla_{\theta_\mu} \mu(\phi(S_j)) \nabla_{A_j} \xi(\phi(S_j), A_j) \tag{16}$$

With continuous iterations, the parameters of the target network are constantly adjusted and updated. Therefore, the parameters of the target network are updated appropriately each time.

$$\theta_{\mu\prime} = \tau\theta_\mu + (1 - \tau)\theta_{\mu\prime} \tag{17}$$

$$\theta_{\xi\prime} = \tau\theta_\xi + (1 - \tau)\theta_{\xi\prime} \tag{18}$$

A model flow chart is shown in Figure 2.

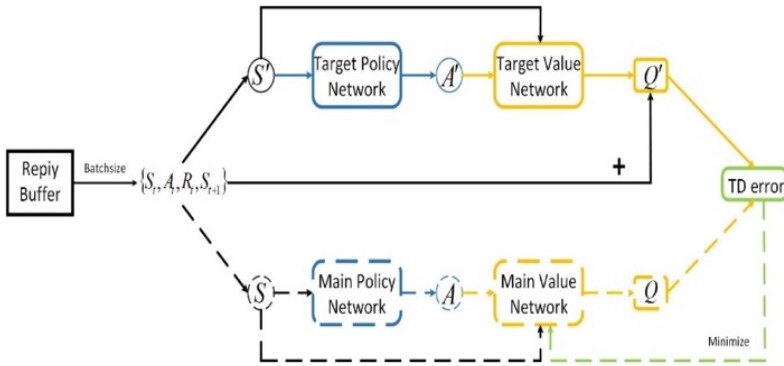

**Figure 2.** DDPG training flow chart.

## 4. Performance Evaluation

MATLAB is utilized for a thorough assessment for the simulation setup and network performance analysis [23].

### 4.1. Simulation Setup

Each time a 2D FOV file is extracted from a film, the user is prompted to choose a different perspective to fully capture motion information. In this article, for simplicity's

sake, each 360° 2D movie is assumed to be a fixed length, i.e., 5 min, and the unit length of the 2D FOV file is taken as 3 Mbit. According to [6], each 360° two-dimensional image has multiple perspectives. In order to ensure accuracy, at least 24 perspectives are used. Each two-dimensional image is segmented 100 times, and each segmentation includes a combination of 24 perspectives to ensure accuracy. There are $M = 720000$ different views. The size of a 3D FOV file is at least twice that of a 2D FOV. Due to the limitation of cache capacity, the overall data volume of an FOV file exceeds the total cache capacity of the server. We analyze the performance of the three schemes compared to a multivideo scenario and present the following conclusions:

(1) CS downloading scheme: After the 3D FOV file is obtained, there are $\mathbf{D}_{1,1}^{n,m} = \mathbf{D}_{1,2}^{n,m} = \mathbf{D}_{1,3}^{n,m} = 1, \forall n \in \mathcal{V}, m \in \mathcal{P}$;

(2) Only caching scheme: The VR device and APs cache the 3D FOV files of all videos based on the 'most popular content (MPC)' rule. VR device and AP denote the numbers of 3D FOV files stored in VR devices and MEC servers: $A_D^{3D} = \frac{Q_D}{\rho}$ and $A_M^{3D} = \frac{KQ_M}{\rho}$. The corresponding numbers of full VR videos stored in the VR devices and MEC servers are denoted as $A_D^V = \frac{A_D^{3D}}{M}$ and $A_M^V = \frac{A_M^{3D} - \left(A_D^{3D} - MA_D^V\right)}{M}$, respectively;

(3) Independent caching and computing scheme: After the 3D FOV file is stored, the video file needs to satisfy the 'MPC' rule. When the device obtains the FOV file from the server, the storage and projection of the 2D FOV file depend on the performance of the MEC server. The number of 2D FOV files is expressed as $A_M^{2D} = Q_D$, and the corresponding number of full VR videos stored on MEC servers is denoted as $A_M^V = \left[\frac{A_M^{2D} - \left(A_D^{3D} - MA_D^V\right)}{M}\right]$.

### 4.2. Simulation Results and Analysis

The impact of $Q_D$ on the mean of the minimal necessary transmission rate (R) for various values and a comparison with alternative methods for $\rho = 2$ are depicted in Figure 3. In the context of limited caching capability, the local hit rate is proportional to $Q_D$, which causes a decrease in the likelihood of retrieving the necessary FOV files from CS and, ultimately, minimizes R. Nevertheless, the size of the 3D FOV file cached on the MEC servers and VR devices in the routes of 3CL and 3CM or sent via the 3CM and 2CCM routes grows with $\rho$, controlling the number of FOV files and ultimately increasing R. Then, R does not change with $Q_D$ due to the use of the CS download scheme with low caching capacity. The reasons for the trend in the cache-only and computation schemes are the same as in the proposed strategy. Furthermore, the proposed strategy has a significantly superior R. This suggests that the cooperation of caching and computing resources between the device and the MEC server can provide benefits to the remaining communication. When the value of $Q_D$ is small, R in the proposed strategy is similar to the independent caching and computing scenario because there are few locally cached FOV files, so the difference in gain is not significant.

A comparison of the influence of K on R in the suggested strategy with other strategies is shown in Figure 4. The value of K can be viewed as the cache capacity of all MEC servers because of the NFV process. When K is in the range of 1 to 5, the cache hit probability of the MEC server increases with an increase in K, and the value of R decreases. When $K > 5$, all data are cached on the server, so R does not change with K. The amount of data cached in the proposed strategy and the independent caching and computing scheme is less than the amount of data stored in the separate cache scheme. Therefore, in the range of $1 \le K \le 7$, R decreases as K increases. Furthermore, when the MEC caching capability varies, the gain from the 2CL-FM and 3CM routes is nearly identical to that of the 2CCM route, making the two curves match to the suggested method and the independent caching and computing scheme.

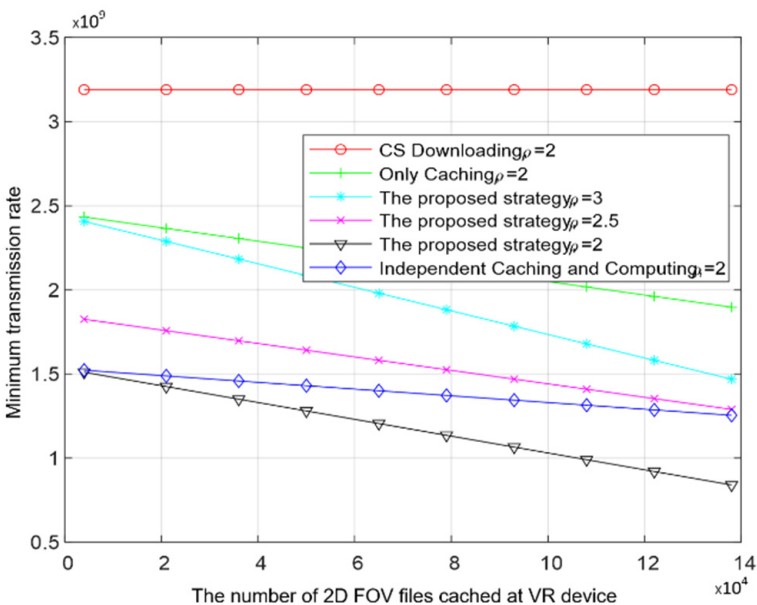

**Figure 3.** The effect of $Q_D$ on $R$ with different $\rho$ values and comparison with other schemes.

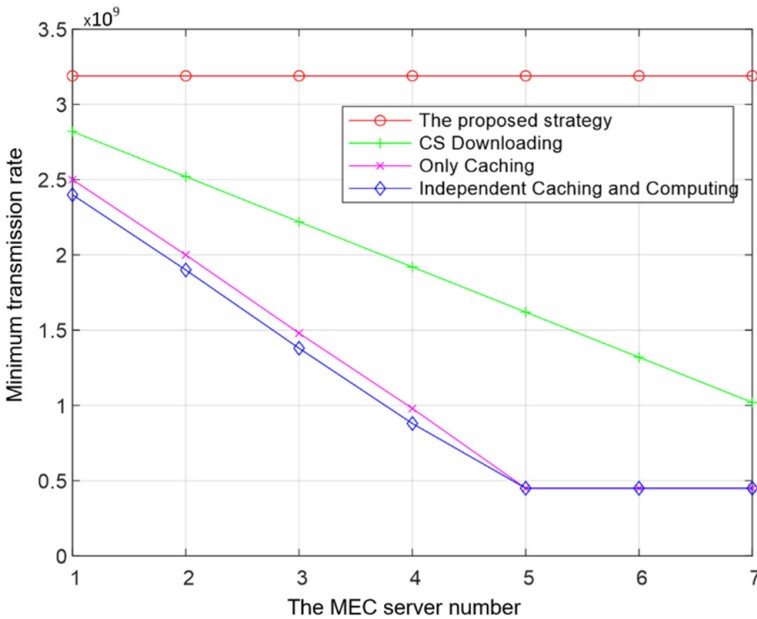

**Figure 4.** The effect of $K$ on $R$ and comparison with other schemes.

The impact of $f_D$ on $R$ at various $P_D$ levels is shown in Figure 5. With the suggested technique, the likelihood of 2CCL and 2CL-FM, or the local computing number, increases with $P_D$, which causes $R$ to decrease. The independent caching and processing method is only used as a benchmark, since the three contrast schemes have no effect on the computing power of the VR device. It goes without saying that the VR device has a significant impact on local computing.

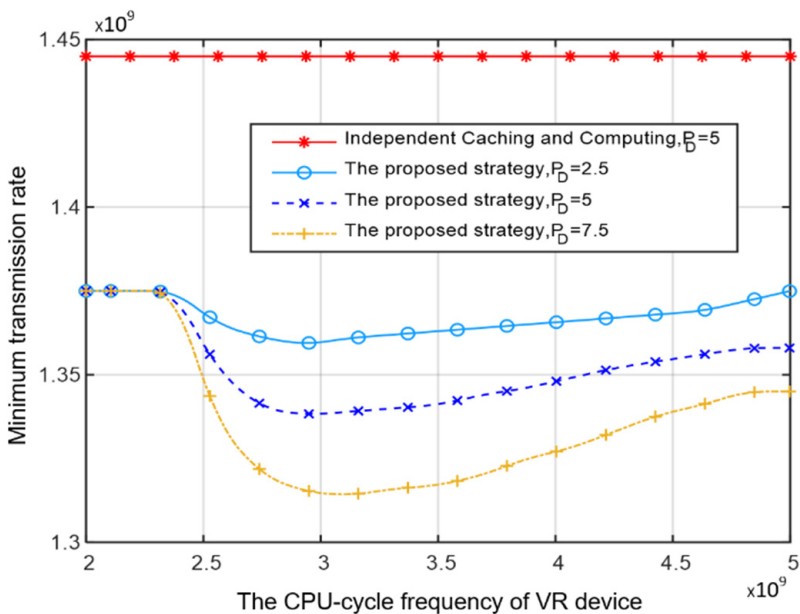

**Figure 5.** The effect of $f_D$ on $R$ with different $P_D$ values.

## 5. Conclusions

This essay is concerned with the VR video industry. A server can precisely compute and store the necessary data for the process of projecting from a 2D FOV to a 3D FOV. In the first section of this work, the service mechanism of VR video was examined; in doing so, an optimization problem for the 3C trade-off was constructed, and a closed equation for the decision matrix was derived. Last but not least, numerical examples were provided to illustrate the value of the recommended method.

**Author Contributions:** Writing—original draft preparation, Y.F.; writing—review and editing, D.W.; validation, Y.H. All authors have read and agreed to the published version of the manuscript.

**Funding:** The work was supported by the National Key R&D Program of China under grant no. 2019YFE0114000.

**Data Availability Statement:** Data are unavailable due to privacy concerns.

**Conflicts of Interest:** The authors declare no conflict of interest.

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
