# Peer review of "Communication, Computing, and Caching Trade-Off in VR Networks"

_electronics, doi:10.3390/electronics12071577_

Round 1
Reviewer 1 Report
Dear Authors,
your article mainly copes with the VR-issues, wich is completely appropriate. However, you mention the term "Digital Twin" in the title and you place this concept in your introduction as well as in your system model in a prominent manner while your elaborations of the digital twin concepts remain on a very high level.
Since the concept relies on a bidirectional 1:1-communication between a physical and a virtual asset/object, the reader gets no clue what you're referring to when you write about a "Digital Twin based Network". There is no single technology for realizing/implementing the digital twin, neither do it have a "fantastic simulation system". It _can_ have a simulation system, but does not need to.
So my proposal is to rewrite, clarify and define the digital twin concepts in more detail and elaborate to which specific concepts of digital twins you refer to. Another possibility is to erase the digital twin concept completely from your article, since in the current form, DTs don't seem to provide any benefit to what could formerly be done without relying to them.
Best regards.
Author Response
Thank you very much for your suggestion, we completely removed the digital twin concept from the article in order to express our point of view more clearly. The title of the article was revised to Communication, Computing and Caching Trade-Off in VR Networks.
Reviewer 2 Report
The manuscript proposes a study aimed at design and developing a digital twin based network focused to satisfy the
transmission requirements of virtual reality video service and improve the quality of the experience.
The paper seems well structured but the concepts expressed are not always clear.
While the description of such activity is clear and the model seems to be properly descibed,
the impact and the added value are poor and shall be better described.
Moreover, as this technique can be computing demanding, related computing resources used and the related performaces have not been mentioned or even described.
Conclusions shall be extended by reporting, as an example, the above mentioned impact and added value.
Personally, I do not believe that the authors did not anticipate further development/experiments of the described model.
So why do not add such future work as part of the conclusions.
Some comments are reported hereafter.
- MATLAB has been adopted for computing calculations and it should be cited in the references.
- Check reference 8. In general, DOI shall appear in the reference.
- Citations have not the same fromat and for some of them year is missing, please check.
Author Response
Thank you very much for your suggestion, we completely removed the digital twin concept from the article in order to express our point of view more clearly. We added relevant reference and checked the format of the reference.
Round 2
Reviewer 1 Report
Dear Authors,
thanks for addressing the issues. I have no further change requests.